# Combined Transcriptomic and Proteomic of *Corynebacterium pseudotuberculosis* Infection in the Spleen of Dairy Goats

**DOI:** 10.3390/ani12233270

**Published:** 2022-11-24

**Authors:** Mingzhe Fu, Xiaolong Xu, Zefang Cheng, Junru Zhu, Ao Sun, Ganggang Xu, Xiaopeng An

**Affiliations:** 1College of Veterinary Medicine, Northwest A&F University, Yangling, Xianyang 712100, China; 2College of Animal Science and Technology, Northwest A&F University, Yangling, Xianyang 712100, China

**Keywords:** *Corynebacterium pseudotuberculosis*, RNA sequencing, proteomic, dairy goat

## Abstract

**Simple Summary:**

Dairy goats, as an important economic livestock for the milk supply in dairy products, have been raised worldwide. However, with the intensive management of dairy goat breeding, the disease problems are becoming increasingly serious, among which *Corynebacterium pseudotuberculosis* has been spreading in dairy goat breeding in China. In this study, we analyzed the transcriptome and proteomics of dairy goats infected with *Corynebacterium pseudotuberculosis* at 72 h and 144 h. Through a comparative correlation analysis of the data, we hope to find the pathogenic mechanism and provide a theoretical basis for the disease problems in dairy goat breeding.

**Abstract:**

*Corynebacterium pseudotuberculosis* (*C. pseudotuberculosis*) is a zoonotic chronic infectious disease. It mainly occurs in dairy goats reared in herds, and once it invades the dairy goats, it is difficult to completely remove it, causing great harm to the development of the sheep industry. This study mainly was based on TMT-based quantitative proteomics and RNA-seq methods to measure the spleen samples of infected dairy goats at different time periods. Nine four-month-old dairy goats were divided into three groups, with three goats in each group. The dairy goats in the first group (NC group) were inoculated with 1.0 mL of sterilized normal saline subcutaneously, and the second (72 h group) and third groups (144 h group) were inoculated with 1.0 mL of 1 × 10^7^ cfu/mL bacterial solution subcutaneously in the neck. Significant changes in the protein and mRNA expression were observed in different infection and control groups. In the 72 h group, 85 genes with differential genes and proteins were up-regulated and 91 genes were down-regulated in this study. In the 144 h group, 38 genes with differential genes and proteins were up-regulated and 51 genes were down-regulated. It was found that 21 differentially expressed genes and proteins were co-up-regulated in the two groups. There were 20 differentially expressed genes and proteins which were co-down-regulated in both groups. The 72 h group were mainly enriched in protein processing in the endoplasmic reticulum, lysosome, amino sugar and nucleotide sugar metabolism and the estrogen signaling pathway. In the 144 h group, they were protein processing in the endoplasmic reticulum pathway which was enriched by mRNA–proteins pairs co-upregulated by the five pairs. The combined transcriptomic and proteomic analyses were performed to provide insights into the effects of *C. pseudotuberculosis* through several regulatory features and pathways. We found that in the early stage of infection (72 h), the co-upregulated gene–protein pairs were enriched in multiple pathways, which jointly defended against a bacterial invasion. However, in the later stages of infection (144 h), when the disease stabilizes, a few co-upregulated gene–protein pairs played a role in protein processing in the endoplasmic reticulum pathway. In addition, the mRNA and protein expressions of dairy goats infected with the bacteria at different periods of time indicated the adaptability of dairy goats to the bacteria. At the same time, it guides us to carry out a corresponding treatment and feeding management for dairy goats according to different periods of time.

## 1. Introduction

Caseous lymphadenitis in sheep and goat (CLA), also known as *pseudotuberculosis* in sheep and goat, is a zoonotic chronic infectious disease caused by *Corynebacterium pseudotuberculosis* [1]. *Corynebacterium pseudotuberculosis* was first isolated by Preisz and Guinand from the kidney abscesses of sheep in 1891 [2]. The disease is distributed throughout the world, and most of the sheep-raising areas in China have the disease. It is recognized as one of the infectious diseases that is most difficult to prevent and control in the world [3].

This disease most commonly infects small ruminants such as goats and sheep, causing significant economic losses in this field [4]. A *Corynebacterium pseudotuberculosis* infection is characterized by the formation of abscesses in superficial lymph nodes (such as parotid and submandibular lymph nodes) or internally (most commonly in lung and mediastinal lymph nodes), causing an inflammatory complex in which inflammatory cytokines are significantly upregulated [5]. Cytokines are the products of activated mononuclear macrophages and are important mediators in the regulation of an infection, inflammation, innate and adaptive immunity. Furthermore, these cytokines have been reported to play an important role in controlling the inflammatory response to bacterial infections [6]. Therefore, *C. pseudotuberculosis* can stimulate a variety of signaling pathways, such as the apoptosis and NF-κB signaling pathways [7]. Currently, the research on *C. pseudotuberculosis* has focused on the isolation and identification of the bacteria, molecular characterization and vaccine development [8]. However, the specific molecular mechanisms underlying the pathogenesis of a *C. pseudotuberculosis* infection are not clear. The immune mechanism of an immune infection in susceptible animals such as dairy goats has not been reported.

*C. pseudotuberculosis* is a kind of polymorphous bacilli, the size of 0.5~0.6 μm × 1.0~3.0 μm, from spherical to rod-shaped, a long bacteria, with one or both ends often expanded into a rod-like, single or into a lattice, plexiform arrangement. It is often rod-shaped in the caseous pus of the lymph nodes. In pure culture smears, most are globular, resembling staphylococcus, and a few are rod shaped. This bacterium does not form capsular and spore, cannot motile, is Gram stain positive and is acid-fast stain negative [1]. This bacterium is a facultative intracellular parasite capable of producing necrotizing, hemolytic exotoxins, the main component of which is phospholipase. The bacteria and *Mycobacterium tuberculosis* have a very similar surface composition and both contain a large amount of lipids [9]. It is thought to resist digestion by phagocytic cells.

In recent years, with the reduction in the cost of and the improvement of technology, high-throughput sequencing has been fully applied in modern agricultural research, mainly in the fields of crop breeding, livestock and poultry disease diagnosis and livestock and the improvement in the poultry breeding guidance, etc., gradually showing its application advantages [8]. Aboshady identified variants in the GIN-sensitive or GIN-resistant creole goat genome by RNA seq, and the T-cell receptor signaling pathway was one of the most important pathways to distinguish the resistant from the susceptible genotypes, with 78% of the genes in this pathway showing a genomic variation. These genomic variants are expected to provide a useful resource for the molecular breeding of GIN resistance in goats [10]. Liu performed a comprehensive transcriptome and proteome analysis of goat skin at 45, 108 and 365 days of age, and found that the KRT and collagen α family proteins may play an important role in the development of the goat hair follicle and wool bending, providing a theoretical basis for revealing the molecular mechanism of goat wool bending [11]. Zhao also revealed that the main biological differences in the casheal fiber diameter differences in Tibetan casheal goats were attributed to intrinsic adaptations related to differences in the metabolism, hypoxia and stress response through the comprehensive study of transcriptome and proteome [12]. In this study, transcriptomics and proteomics were used to jointly analyze the spleen tissues of dairy goats infected with *C. pseudotuberculosis* at different time periods, mainly to clarify the mRNA and protein expression in the spleen tissues of dairy goats at different time periods after an infection with *C. pseudotuberculosis*. This paper attempts to explain the intrinsic relationship between the mRNA and protein expression, so as to provide theoretical basis for the molecular breeding of dairy goats.

## 2. Materials and Methods

### 2.1. Ethics Statement and Animals

This study was approved by the Institutional Animal Care and Use Committee of Northwest A&F University (permit number: 17-347, data: 2017-10-13) following the recommendation of the Regulations for the Administration of Affairs Concerning Experimental of China. Additionally, the healthy dairy goats were from Shaanxi Province, China. The 4-month-old experimental dairy goats, weighing about 25 kg, were raised individually and fed a normal diet in the laboratory animal room.

### 2.2. Schematic Representation of Workflow

This study illuminated to characterize the differences between 72 h and 144 h infected dairy goats with *C. pseudotuberculosis* by TMT proteomics-based quantitative technology and RNA sequencing technology. The experimental design and workflow is shown in Figure 1. Additionally, the dairy goats were randomly divided into three groups. All three groups (*n* = 3) were injected subcutaneously into the neck. The first group was given 1mL of normal saline as the control. The second group was inoculated with 1 mL of 10^7^ cfu/mL live *C. pseudotuberculosis* for 72 h. The third group was inoculated with 1 mL of 10^7^ cfu/mL live *C. pseudotuberculosis* for 144 h.

### 2.3. Collection Spleen Samples of Dairy Goats

In this study, the temperature and symptoms of the dairy goats were monitored regularly every day. On the second day of the infection, the dairy goats showed symptoms of depression and anorexia, and their body temperature kept rising. On the third day of the infection, there was a significant swelling at the vaccination site in the neck. On the 6th day of the infection, the swelling became more pronounced with a pustular rupture. At this point, the spleen tissues were collected and weighed at the 72nd hour of the infection in the second group (on the 3rd day) and at hour 144 of the infection in the first (control group) and third groups (on the 3rd day). Then, the spleens were washed in ice cold NaCl solution (0.9%) and snap-frozen in liquid nitrogen. In the end, the spleen samples of the dairy goats were stored at −80 °C.

### 2.4. RNA Sequencing

This analysis was performed as previously described [13]. Briefly, three spleen samples from each group were included for RNA sequencing. The total RNA was extracted by the mirVana^TM^ miRNA Isolation Kit (Ambion, Austin, TX, USA) and stored at −80 °C. Meanwhile, the RNA quality was analyzed using the NanoDrop 2000 (Thermo, Waltham, MA, USA). Additionally, the cDNA library was performed by the TruSeq RNA Library Prep Kit v2. After extracting the total RNA and digesting the DNA with DNase, magnetic bead rich mRNA with Oligo (dT) was used. This was done by adding the interrupted reagent to break the mRNA into short fragments, using the interrupted mRNA as the template, using six-base random primers to synthesize the one-strand cDNA, then preparing the two-strand synthesis reaction system to synthesize the two-strand cDNA and using the kit to purify the double-strand cDNA. The purified double-stranded cDNA was then repaired, a-tailed and sequenced, followed by a fragment size selection and PCR amplification. The constructed libraries were evaluated by the Agilent 2100 bioanalyzer (Santa Clara, CA, USA). Finally, RNA-seq was performed on an Illumina HiSeq^TM^ 2500, and it generated 150 bp paired-end reads.

### 2.5. Proteomic Analysis

A total of 9 spleen samples were prepared for the proteomic analysis. This analysis was performed as previously described [14]. Briefly, a piece of splenic tissue was added to the centrifuge tube. Lysates were then added, crushed for 3 min and centrifuged three times at room temperature. A total of 20 μg of the samples were subjected to SDS-PAGE. The separated gels were stained with Coomassie Brilliant Blue R-250 (GE Healthcare, Beijing, China) and scanned with an image scanner (GE Healthcare). Reducing buffer was added to each sample, 100 μg of protein were added and incubated at 60 °C for 1 h, then the IAA was adjusted to the appropriate final concentration (50 mM) and the reaction was performed in the dark for 40 min. The samples were centrifuged at 12,000 RPM for 20 min. TEAB buffer was added and centrifuged for 20 min. Subsequently, the column was placed into a new tube and incubated at 37 °C for 12 h, followed by a final centrifugation for 20 min to collect the peptides. TEAB buffer was added to the tubes and centrifuged for half an hour, and the tube bottom was collected and freeze-dried. In addition, TEAB buffer was added to the samples. An amount of 40 μL was used for the labeling reaction. In the end, the peptide mixture was entered into Q-ExActivems (Thermo Fisher Technology Inc., Waltham, MA, USA) for LC-MS/MS analysis.

### 2.6. Bioinformatics Analysis

In this study, the Trimmomatic software (version 0.36) [15] was used for the quality control of raw reads, and the read data containing the poly-N, adapters and the low-quality reads were deleted. The Q30 and GC contents of the clean reads were also calculated to evaluate the quality of the reads. Hisat2 software (2.2.1.0) [16] was used to map clean reads to the Capra hircus reference genome to obtain the location information and sequence characteristics of the clean reads. A sequence similarity alignment was used to identify the expression abundance of each protein coding gene in each sample. Htseq-count software (version 0.9.1) [17] was used to obtain the number of reads aligned to the protein-coding genes in each sample, and CuffLink software (version 2.2.1) [18] was used to calculate the fragments per kilobase of the exon model per million mapped fragments (FPKM) value of the protein-coding genes expression. DESeq software (version 1.18.0) [19] was used to standardize the counts of each sample gene and calculated the difference fold, and NB (negative binomial distribution test) was used to test the significance of the difference in the number of reads. Finally, the differential protein-coding genes were screened according to the fold difference and significant difference test results. The selection criteria were, first, the FoldChange, which is the multiple change in the expression level of the same gene in two samples, and the second was the adjusted *p*-value (FDR). The calculation method of the FDR value should first calculate the *p*-value of each gene, and then use the FDR the error control method to adjust the *p*-value by multiple hypothesis testing. The default filter for the differences were a *p*-value < 0.05 and the FoldChange > 2.

The experimental data of the proteomic were analyzed by Proteome DiscovererTM 2.3 (Thermo Company, Waltham, MA, USA) software, and the database used was the Uniprot goat database (https://www.uniprot.org) accessed on 12 July 2019. The false positive rate of the identification of peptides was less than 1%. The specific database search parameters are set as follows: the sample type: TMT 10 plex (peptide labeled), Cys. alkylation: iodoacetamide, digestion: trypsin, instrument: Q Exactive HF, database: uniport-proteome_UP00291000-*Capra hircus* (goat). Then, the retrieved results were obtained according to a Score Sequest HT > 0 and a unique peptide ≥ 1, and the blank values were removed. Based on the selected credible proteins, the FoldChange and the *p*-value of the difference significance of the comparison group were calculated, and the FoldChange > 1.2 and a *p*-value < 0.05 was the standard for screening the significant difference proteins.

## 3. Results

### 3.1. Combined Results of Differential Proteins and Differential Genes

First, the transcriptomic data and proteomic data of the goats infected with *C. pseudotuberculosis* for 72 h and 144 h were jointly analyzed according to the log2 values of the fold differences in the protein and gene expression (Figure 2). In the 72 h group, 85 genes with differential genes and proteins were up-regulated and 91 genes were down-regulated in this study (Appendix A). Interestingly, seven differentially expressed genes were up-regulated and differentially expressed proteins were down-regulated, and one differentially expressed gene was down-regulated and differentially expressed proteins were up-regulated. In the 144 h group, 38 genes with differential genes and proteins were up-regulated and 51 genes were down-regulated in this study (Appendix A). Among them, the expression of one differential gene was up-regulated and the expression of the differential protein was down-regulated, and the expression of the two differential genes was down-regulated and the expression of the differential protein was up-regulated. Then, the intersection parts of the differential genes and differential proteins in the 72 h group and 144 h group were taken in this study, and the genes and proteins used in the transcription proteins were the parts that were up-regulated and down-regulated in the 72 h group and 144 h group. It was found that 21 differentially expressed genes and proteins were co-up-regulated in the two groups. Additionally, there were 20 differentially expressed genes and proteins were co-down-regulated in both groups (Appendix A).

### 3.2. Spearman Correlation Analysis

In this study, Spearman correlation analysis was performed on the obtained transcriptome and proteome data with the same trend, and it was found that 56.62% of the mRNA–protein pairs in the 72 h group had a positive correlation and 6.59% had a significant positive correlation (Appendix A) (Figure 3A). in the 144 h group, 56.5% of the mRNA–protein pairs had a positive correlation and 4.38% had a significant positive correlation. Among the groups with an intersection at 72 h and 144 h, 97.5% of the 41 mRNA–protein pairs had a positive correlation and 45% had a significant positive correlation (Appendix A) (Figure 3B).

### 3.3. Gene Ontology and Kyoto Encyclopedia of Genes and Genomes Analysis

In this study, enrichment analysis was carried out for the co-up-regulated mRNA and protein genes in the 72 h group and the 144 h group. First, through GO (Gene Ontology) analysis (Figure 4), it was found that the top 30 terms selected in the 72 h group in this study are mainly enriched in an epithelial cell differentiation and protein glycosylation, and the response to it is mainly enriched in the biological process. The endoplasmic reticulum lumen, endoplasmic reticulum chaperone complex and spindle microtubule are in the cellular component, and the scavenger receptor activity, microtubule motor activity and ATPase activity are in the molecular function (Appendix A). Within the 144 h group, the regulation of the immune response, complement activation, classical pathway and receptor-mediated endocytosis is in the biological process; the extracellular space, collagen trimer and extracellular region is in the cellular component; and the scavenger receptor activity, unfolded protein binding and calcium ion binding is in the molecular function (Appendix A). At the same time, this study also found that both groups were highly enriched in the scavenger receptor activity, indicating that this function has been playing a key role.

In this study, KEGG (the Kyoto Encyclopedia of Genes and Genomes) enrichment analysis was used to elucidate the signal transduction pathways mainly involved in the jointly upregulated mRNA–protein pairs (Figure 5). The results were as follows: in the 72 h infection group, four pathways were enriched by 18 co-upregulated mRNA–protein pairs. They were mainly enriched in the protein processing the endoplasmic reticulum, lysosome, amino sugar and nucleotide sugar metabolism and estrogen signaling pathway (Appendix A). In the 144 h infection group, the protein processing in the endoplasmic reticulum pathway was enriched by mRNA–protein pairs which were co-upregulated by the five pairs (Appendix A).

### 3.4. Expression Cluster Analysis

Unsupervised hierarchical clustering was performed on the expression levels of the corresponding genes/proteins in the two omics. Generally speaking, the same type of genes/proteins can appear in the same cluster, and the genes clustered in the same cluster may have similar biological functions. Then, when the intersection parts of the differential genes and differential proteins in the 72 h group and 144 h group were taken in this study, we found that there were 21 differentially expressed genes and the proteins were co-up-regulated and there were 20 differentially expressed genes and these proteins were co-down-regulated in both groups. It is clear from the heatmap (Figure 6) that the 41 gene–protein pairs were significantly different between the infected and control groups. Among them, 21 gene–protein pairs were highly expressed in the infection group and lowly expressed in the control group. The expression of 20 gene–protein pairs was low in the infected group and high in the control group. Among them, genes such as ALPL, FABP3, VCAN and SLC11A1 have been highly expressed at different times and ALOX15, SNCG, EPDR1 and CNN1 were all down-regulated compared with normal dairy goats.

## 4. Discussion

In the present study, the transcriptomic and proteomic analyses of goat spleen tissue revealed the combined effects of mRNA–protein pairs during a *C. pseudotuberculosis* infection at different times. It is a chronic infectious disease that can cause zoonosis. This bacterium can cause disease in sheep, goats, cattle and other animals [20]. Nodules of varying sizes can be observed in the spleen, containing yellowish case-like substances. Although the disease has a slow onset and low fatality rate, it can also lead to an emaciation and decreased production performance of the diseased sheep. Additionally, after the onset of antibiotics treatment, the effect is not good [21]. In this study, dairy goats were inoculated subcutaneously with 10^7^ cfu/mL of *C. pseudotuberculosis* solution and were compared with normal dairy goats to discover the pathogenic mechanism as far as possible.

In the 72 h infection group and the 144 h infection group, different mechanisms played different roles in the spleen over time after an infection with *C. pseudotuberculosis*. Our results show that there are clear differences between the mRNA and protein in *C. pseudotuberculosis*-infected spleen tissues at different time periods, but we also find some consistent effects at different time periods. With the identification of *C. pseudotuberculosis*, as a splenic infection is consistent with our previous studies, the effect of this bacterium is related to the passage of time. According to our transcriptomic and proteomic studies, an early infection may trigger immune-related pathways, and the molecular mechanism can be more fully observed in the 144 h infection group. In conclusion, this study speculated that these genes must play a key role in the process of a *C. pseudotuberculosis* infection in dairy goats. We focused on the protein and mRNA level of enrichment analysis of differentially expressed genes at the same time and found that the gene expression mainly in the basal metabolic pathways related to the immune pathways. Interestingly, we also found that in the two groups were sick genes mostly enriched in the basal metabolic pathway; we suspect that this is because the *C. pseudotuberculosis* fatality rate is low and has a slow onset. First of all, the GO items such as the epithelial cell differentiation, protein glycosylation, spindle microtubule, endoplasmic reticulum chaperone complex and kinesin complex, scavenger receptor activity and microtubule motor activity, etc. All these data suggested that the bacteria might cause basic metabolic diseases by infecting dairy goats, which may induce the decline in immune ability and further induce diseases. Through KEGG analysis, the genes were mainly enriched in the protein processing in the endoplasmic reticulum, lysosome, amino sugar and nucleotide sugar metabolism and estrogen signaling pathway. This further suggested that the disease caused by the bacteria is a chronic disease, which induced the low immune ability of the body. Among them, the ALPL, FABP3, VCAN and SLC11A1 genes have been highly expressed at different times and the ALOX15, SNCG, EPDR1 and CNN1 genes were all down-regulated compared with normal dairy goats. 

At the same time, this study also found some interesting problems. When we analyzed the data, we found that the mRNA level and protein expression level of some of the data were opposite. In the 72 h infection group, the mRNA expression was up-regulated and the protein expression was down-regulated in seven groups, namely the TNXB, NFIA, PROC, SPIN1, NR2C2 nuclear receptor subfamily 2 group C member 2, Carboxypeptidase D and ARHGAP11B. TNXB depletion prolonged the corneal epithelial wound healing and increased the neutrophil inflammation in response to debridement in mice. TNXB also helped control the neutrophil infiltration to support regenerative responses to injury and prevent oxidative stress mediators from rising to cytotoxic levels [22].

NFIA differentially controls adipogenic and myogenic gene programs through different pathways to ensure a brown and beige adipocyte differentiation [23]. The RP5-833A20.1/HSA-Mir-382-5p/NFIA pathway is critical for regulating cholesterol homeostasis and the inflammatory response in human acute monocytic leukemia macrophages [24]. PROC is a multifunctional serine protease with anticoagulant, cytoprotective and anti-inflammatory activities. In addition to the cytoprotective effects of PROC on endothelial cells, podocytes and neurons, PROC also cleaves and detoxifies extracellular histones, which are major components of neutrophil extracellular traps (NET) [25]. PROC is a natural anticoagulant that exerts anti-inflammatory and cytoprotective effects in various diseases through pathways mediated by the endothelial protein C receptor (EPCR) and protease-activated receptor (PAR) [26]. The PROC played an active role in preventing the progression of many disease processes [27]. TGF-β-activated kinase 1 (TAK1) is a key regulator involved in different innate immune signaling pathways. EHEC Tir negatively regulates proinflammatory responses by inhibiting the activation of TAK1, which is essential for an immune evasion and may be a potential target for the treatment of bacterial infections [28]. LYTAK1 may inhibit the LPS-induced production of several proinflammatory cytokines and endotoxic shock by blocking TAK1-regulated signaling [29]. Transforming the growth factor-β (TGF-β) is a multifunctional cytokine, mainly as an inhibitor of the immune function [30]. Meanwhile, the mRNA expression of TNFRSF13C was down-regulated and the protein expression was up-regulated in the 72 h infection group. The B-cell activator of the TNF family receptor (BAFF-R) encoded by the TNFRSF13C gene is essential for the survival of transitional B cells to maturity. BAFF-R exerts a signaling function by inducing the activation of NF-κappaB [31]. Finally, we also found three sets of data with opposite mRNA and protein expression trends in the 144 h group, among which the CTGF gene is of interest. It is a multifunctional protein in the CCN family that regulates the cell proliferation, differentiation, adhesion and various other biological processes. It is involved in disease-related pathways such as the Hippo pathway, p53 and nuclear factor Kappa-B (NF-κB) pathway, and thus contributes as a downstream effector to the development of inflammation, fibrosis, cancer and other diseases [32].

Most of the relations between the RNA and protein were found in the 72 h reinfection group. We speculate that *C. pseudotuberculosis* may enter the goat first, which leads to the involvement of multiple biological pathways in the goat. With the passage of time, some previous biological pathways play a role, meaning the internal environment tends to be stable, as well as the disease. As a result, some metabolic pathways in the body adapted to the invasion of bacteria, and only a part of the biological pathways continued to play a role.

## 5. Conclusions

In this study, we demonstrated that the infection of dairy goats with *C. pseudotuberculosis* at different time periods resulted in different mRNA and protein expressions. We found that in the early stage of the infection (72 h), co-upregulated gene–protein pairs were enriched in multiple pathways, which jointly defended against a bacterial invasion. Among them, genes such as ALPL, FABP3, VCAN and SLC11A1 have been highly expressed at different times and ALOX15, SNCG, EPDR1 and CNN1 were all down-regulated compared with normal dairy goats. At the same time, this study also found some interesting problems. When we analyzed the data, we found that the mRNA level and protein expression level of some data were opposite. In the 72 h infection group, the mRNA expression was up-regulated and the protein expression was down-regulated in 7 groups, namely the TNXB, NFIA, PROC, SPIN1, NR2C2 nuclear receptor subfamily 2 group C member 2, Carboxypeptidase D and ARHGAP11B. This phenomenon may be related to the epigenetic modification of the mRNA and protein. However, in the later stages of the infection (144 h), when the disease stabilizes, a few co-upregulated gene–protein pairs play a role in protein processing in endoplasmic reticulum pathway. In addition, we also proved that the different times of a *C. pseudotuberculosis* infection with different RNA and protein expressions, and the related pathways, produce different effects. This emphasis on dairy goats with a *C. pseudotuberculosis* infection over different time periods corresponds to the treatment and raising management of dairy goats.

## Figures and Tables

**Figure 1 animals-12-03270-f001:**
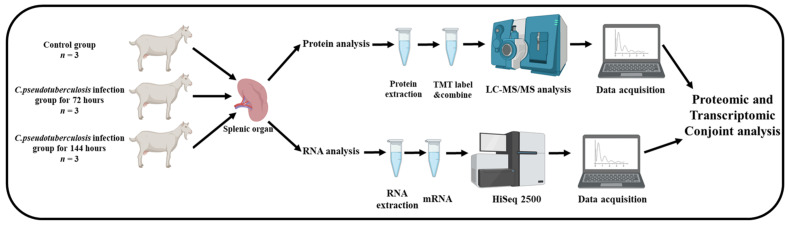
Experimental design and workflow of RNA sequencing and proteomic comparison of dairy goats.

**Figure 2 animals-12-03270-f002:**
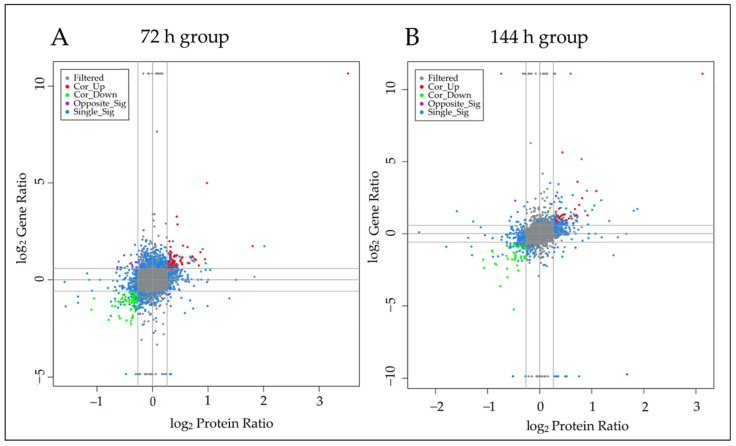
Comparison of mRNA and protein changes in infected dairy goats in different time. (**A**) is a comparison of mRNA and protein in the 72 h infection group. (**B**) is a comparison of mRNA and protein in the 144 h infection group.

**Figure 3 animals-12-03270-f003:**
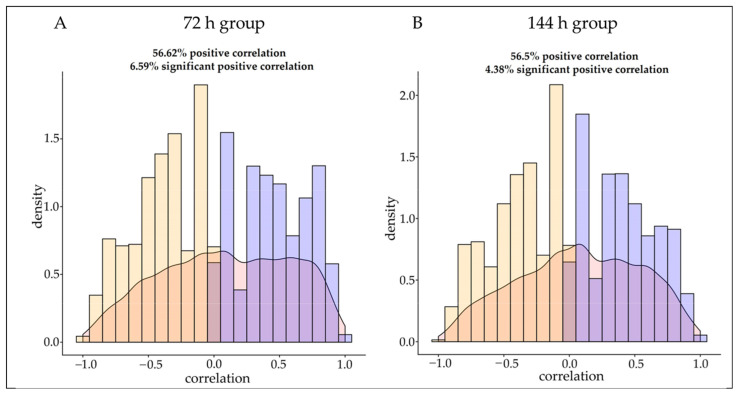
Correlation between protein and mRNA abundance in infected dairy goats. Density plot (**A**) showing the global Spearman correlation for protein–mRNA pairs with the 72 h infection group. Density plot (**B**) showing the global Spearman correlation for protein–mRNA pairs with the 144 h infection group.

**Figure 4 animals-12-03270-f004:**
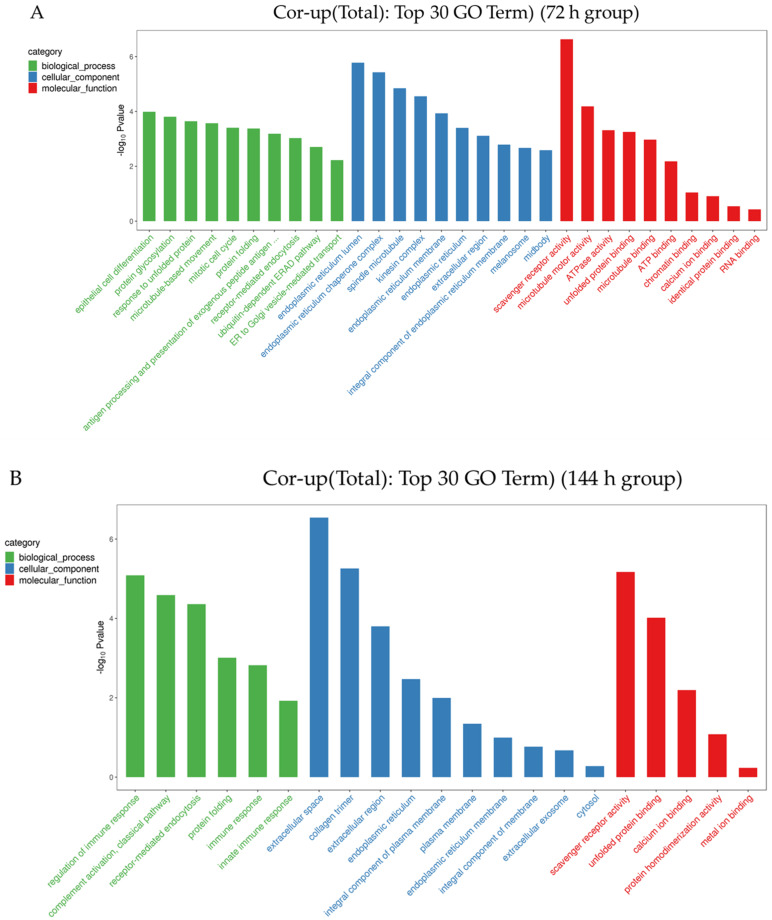
The enriched GO (gene ontology) terms concluding the biological process, molecular function and cellular component of co-upregulated proteins and transcriptome in different groups ((**A**) means 72 h, (**B**) means 144 h).

**Figure 5 animals-12-03270-f005:**
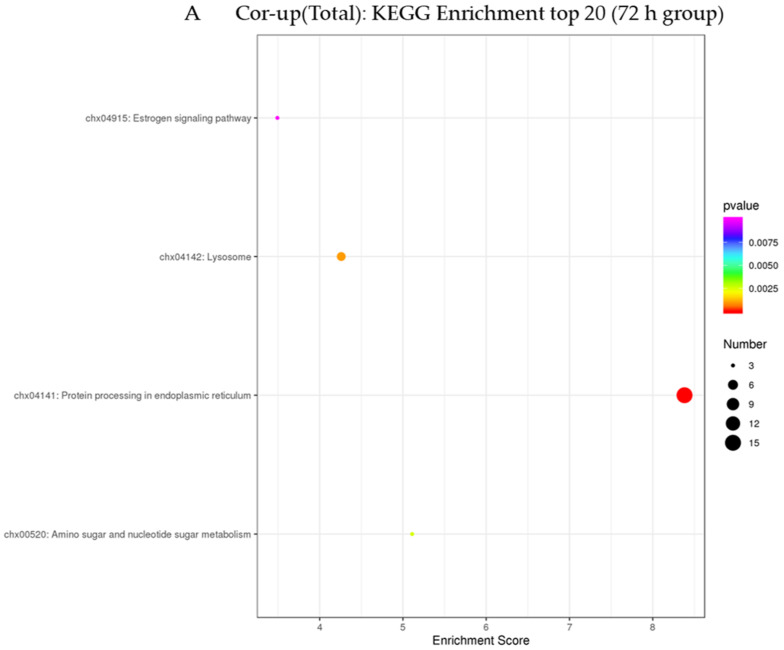
KEGG (the Kyoto Encyclopedia of Genes and Genomes) pathway classification and functional enrichment for the predicted co-upregulated proteins and transcriptome in different groups ((**A**) means 72 h, (**B**) means 144 h).

**Figure 6 animals-12-03270-f006:**
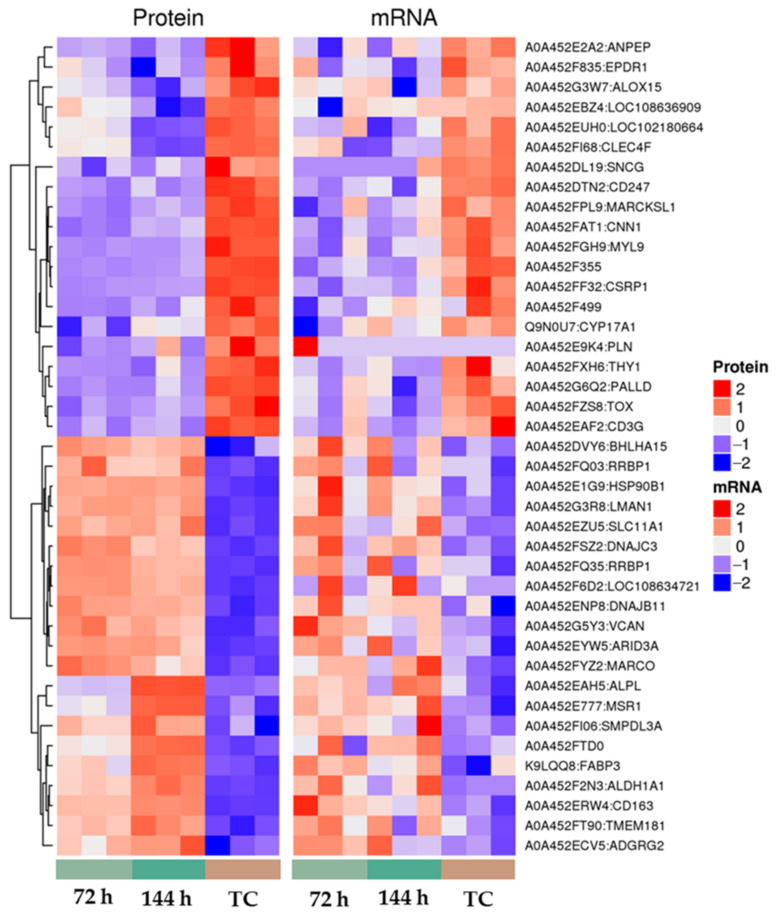
Transcriptomes and proteins co-expressed in infected groups at different time periods (a total of 41 mRNA–protein pairs were co-expressed).

## Data Availability

All relevant information is provided in this manuscript.

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
