# Peer review of "Combined Transcriptomic and Proteomic of Corynebacterium pseudotuberculosis Infection in the Spleen of Dairy Goats"

_animals, 2022, doi:10.3390/ani12233270_

Round 1
Reviewer 1 Report
The following are the specific questions of the article:
1. What is the significance of this result study that should be stated in the abstract.
2. There are many repetitive statements in the first and second paragraphs of the introduction, which can be appropriately removed.
3. Some examples of the application of transcriptomics and proteomics in animal husbandry can be added in the last paragraph of the introduction.
4. In the discussion, you can delete that sentence "Combined analysis showed that the gene was upregulated at both mRNA and protein levels after infection with C. tuberculosis. ", to make the whole sentence smooth.
5. Whether some of the specific genes found are indicated in the second paragraph of the discussion.
6. 6. For RNA-seq and Protein-seq, How to build the library was not clarified, especially the application of samples. Which should be presented in Materials and Methods.
Author Response
Reviewer 1:
1 What is the significance of this result study that should be stated in the abstract.
Explanation: This article mainly wants to sample the spleen of dairy goats at different times, explain the pathogenic mechanism through sequencing data, and provide theoretical guidance for production.
In the article, we added this paragraph to show the significance of our results ---- We found that in the early stage of infection (72h), co-upregulated gene-protein pairs were enriched in multiple pathways, which jointly defended against bacterial invasion. However, in the later stages of infection (144h), when the disease stabilizes, a few co-upregulated gene-protein pairs play a role in protein processing in endoplasmic reticulum pathway. In addition, the mRNA and protein expressions of dairy goats infected with the bacteria at different periods of time indicated the adaptability of dairy goats to the bacteria. At the same time, it guides us to carry out corresponding treatment and feeding management for dairy goats according to different periods of time.
2 There are many repetitive statements in the first and second paragraphs of the introduction, which can be appropriately removed.
Correction: We thank the reviewer for your suggestions that made our article clearer. In the introduction of the article, we cut these two paragraphs ---- “Once it invades the sheep, it is difficult to completely remove, causing great harm to the development of sheep raising industry.” and “Corynebacterium pseudotuberculosis is a zoonotic animal pathogen that causes caseous lymphadenitis in sheep and goats.”
3 Some examples of the application of transcriptomics and proteomics in animal husbandry can be added in the last paragraph of the introduction.
Correction: We thank the reviewer for your suggestions to enrich our article. In the article, we added these paragraphs ---- “Aboshady identified variants in the Gin-sensitive or Gin-resistant Creole goat genome by RNA seq, and the T-cell receptor signaling pathway was one of the most important pathways to distinguish resistant from susceptible genotypes, with 78% of the genes in this pathway showing genomic variation. These genomic variants are expected to pro-vide a useful resource for molecular breeding of GIN resistance in goats. [10] Liu per-formed a comprehensive transcriptome and proteome analysis of goat skin at 45, 108, and 365 days of age, and found that KRT and collagen α family proteins may play an important role in goat hair follicle development and wool bending, providing a theo-retical basis for revealing the molecular mechanism of goat wool bending [11] zhao also revealed that the main biological differences in the casheal fiber diameter differ-ences of Tibetan casheal goats were attributed to intrinsic adaptations related to dif-ferences in metabolism, hypoxia, and stress response through the comprehensive study of transcriptome and proteome. [12]”
4 In the discussion, you can delete that sentence "Combined analysis showed that the gene was upregulated at both mRNA and protein levels after infection with C. tuberculosis. ", to make the whole sentence smooth.
Correction: We thank the reviewer for your suggestions. We have removed this sentence from the discussion section
5 Whether some of the specific genes found are indicated in the second paragraph of the discussion.
Correction: We thank the reviewer for your suggestions. And we have included examples of related genes in the discussion of this article. The specific sentences are as follows ---- “Among them, ALPL, FABP3, VCAN, SLC11A1 genes have been highly expressed at different times and ALOX15, SNCG, EPDR1, CNN1 genes were all down-regulated compared with normal dairy goats.”
6 For RNA-seq and Protein-seq, how to build the library was not clarified, especially the application of samples. Which should be presented in Materials and Methods.
Thanks to the reviewer for comments that made our approach clear. In previous articles, we have described the library construction of RNA-seq and protein-seq in detail, but it is our mistake not to quote the previous article. If some methods are explained again in this article, they will be repeated in the previous article. we hoped the reviewer understand.
Reviewer 2 Report
Paragrap 2.3: when the spleen was removed? At lines 118-119 you wrote "on day 6th of infection, swelling became more pronounced with postular rupture. At this point, the spleen was removed and weighed".
Paragraph 2.4: pay attention at puntuaction (line 132).
Paragraph 2..5: pay attention at capital letter (line139)
Paragraph 2.6: fix "ploy-N" to "poly-N" (line 176).
Paragraph 3.3: pay attention at puntuaction (from line 250 to 257)
Paragraph 3.4: pay attention at capital letter (line 286)
Chapter 4: pay attention at puntuaction (line 351) and pay attention for italics (line 363)
Author Response
Reviewer 2:
Paragraph 2.3: when the spleen was removed? At lines 118-119 you wrote "on day 6th of infection, swelling became more pronounced with postular rupture. At this point, the spleen was removed and weighed".
Explanation: We thank the reviewer for your suggestions. Meanwhile, we explained in the article that spleen tissues were collected at 72h of infection in the second group (on day 3th) and at 144h of infection in the first (control group) and third groups (on day 3th).
Paragraph 2.4: pay attention at puntuaction (line 132).
Correction: The reviewer was greatly appreciated for your suggestions on the details of this article. We have made changes in the article
Paragraph 2.5: pay attention at capital letter (line139).
Correction: The reviewer was greatly appreciated for your suggestions on the details of this article. We have made changes in the article. The “A” was changed to “a”
Paragraph 2.6: fix "ploy-N" to "poly-N" (line 176).
Correction: The reviewer was greatly appreciated for your suggestions on the details of this article. We have made changes in the article.
Paragraph 3.3: pay attention at puntuaction (from line 250 to 257)
Correction: The reviewer was greatly appreciated for your suggestions on the details of this article. We have made changes in the article.
Paragraph 3.4: pay attention at capital letter (line 286)
Correction: The reviewer was greatly appreciated for your suggestions on the details of this article. We have made changes in the article.
Chapter 4: pay attention at puntuaction (line 351) and pay attention for italics (line 363)
Correction: The reviewer was greatly appreciated for your suggestions on the details of this article. We have made changes in the article. Also, we have replaced italics throughout the article.
Reviewer 3 Report
Article aims with different gene regulation during infection of dairy goats with C. pseudotuberculosis. Results of experiments should explain impact of infection on changes in mRNA expression and protein synthesis in two different time after infection. Autors apply very informative and modern methodology of combined transcriptomic and proteomic analyses with possibility of revealing of huge number of important results, relevant for the field and presented in a well-structured manner.
In the manuscript issue of influence of infection with C. pseudotuberculosis to mRNA expression and protein sythesis is explained in introduction section clearly and with relevant number of references. Materials used in the study were chosen adequately for achieving results for comparison between two examined group and negative controle. Applied methodology is the methods are appropriate for the set hypothesis
and presented in a well-structured manner. Results were shown clearly with extensively comments in discussion section and enough number of figures.
Authors should improve:
Abstract section contain very clear explanations of the aim, and results but abstract section doesn’t have explanation of methodology, methods were not mentioned. Above that, there is no conclusion. The last sentence “Combined transcriptomic and proteomic analyses were performed to provide insights into the effects of C.pseudotuberculosis through several regulatory features and pathways” looks like the aim of article.In the Results authors should highlight the names of the genes that showed the highest degree of differences in mRNA expression an proteins compared to the control.
Discussion is confused and need major revision with more references and comparison with similar investigation of other bacterial infections.
In reference section you should adopt and correct references according to the rules of the journal Animals, like this:
“Mota, P.C.; Ehmcke, J.; Westernströer, B.; Gassei, K.; Ramalho-Santos, J.; Schlatt, S. Effects of different storage protocols on cat testis tissue potential for xenografting and recovery of spermatogenesis. Theriogenology 2012, 77, 299–310.”
Conclusion need major revision with referring to specific genes with the most differences found in experiments.
Specific comments
- You could name three testing group with symbols, for example: group A (72 hours), group B (144 hours) and group C injected with normal saline (control).
- Corynebacterium pseudotuberculosis or C. pseudotuberculosis should be in italic as Latin species name in every part of article.
- “and most of the sheep raising areas in our country have the disease”, please cite the name of your country.
In Materials and Methods section, In Figure 1. You wrote C. Pseudotuberculosis with uppercase instead of C. pseudotuberculosis
Author Response
Reviewer 3:
Abstract section contained very clear explanations of the aim, and results but abstract section doesn’t have explanation of methodology, methods were not mentioned. Above that, there is no conclusion. The last sentence “Combined transcriptomic and proteomic analyses were performed to provide insights into the effects of C.pseudotuberculosis through several regulatory features and pathways” looks like the aim of article.
Correction: This article mainly wants to sample the spleen of dairy goats at different times, explain the pathogenic mechanism through sequencing data, and provide theoretical guidance for production. And we restated the method.
In the article, we added this paragraph to show the significance of our results ---- We found that in the early stage of infection (72h), co-upregulated gene-protein pairs were enriched in multiple pathways, which jointly defended against bacterial invasion. However, in the later stages of infection (144h), when the disease stabilizes, a few co-upregulated gene-protein pairs play a role in protein processing in endoplasmic reticulum pathway. In addition, the mRNA and protein expressions of dairy goats infected with the bacteria at different periods of time indicated the adaptability of dairy goats to the bacteria. At the same time, it guides us to carry out corresponding treatment and feeding management for dairy goats according to different periods of time.
And the presentation of the method in the abstract is modified to follow ---- This study mainly based on TMT-based quantitative proteomics and RNA-seq methods to measure spleen samples recorded and compared the changes of protein and transcriptome in infected of dairy goats at different time periods. Nine four-month-old dairy goats were divided into three groups with 3 goats in each group. Dairy goats in the first group (NC group) were inoculated with 1.0 mL sterilized normal saline subcutaneously, and in the second group (72h group) and third groups(144h group) were inoculated with 1.0 mL 1×107 cfu/mL bacterial solution subcutaneously in the neck. Dairy goats in the first and second (experimental group) groups were inoculated with 1.0 mL 1×107 cfu/mL bacterial solution subcutaneously in the neck, and in the third group (control group) were inoculated with 1.0 mL sterilized normal saline subcutaneously in the neck. Spleen tissue samples were taken at 72 hours and 144 hours.
In the Results authors should highlight the names of the genes that showed the highest degree of differences in mRNA expression a protein compared to the control.
Correction: We thank the reviewer for your suggestions. We have made changes in the article. The 21 gene-protein pairs were highly expressed in the infection group and low expressed in the control group. The expression of 20 gene-protein pairs was low in the infected group and high in the control group. Among them, genes such as ALPL, FABP3, VCAN and SLC11A1 have been highly expressed at different times and ALOX15, SNCG, EPDR1, CNN1 were all down-regulated compared with normal dairy goats. At the same time, this study also found some interesting problems. When we analyzed the data, we found that the mRNA level and protein expression level of some data were opposite. In the 72h infection group, mRNA expression was up-regulated and protein expression was down-regulated in 7 groups, TNXB, NFIA, PROC, SPIN1, NR2C2 nuclear receptor subfamily 2 group C member 2, Carboxypeptidase D, ARHGAP11B were included.
Discussion is confused and need major revision with more references and comparison with similar investigation of other bacterial infections.
Explanation: Thank you for the reviewer's suggestions on the discussion of this article. We have made a part of the modification of the discussion. In the discussion, we mainly introduced the problems caused by the infection of milk goats with C.pseudotuberculosis, then discussed according to the results, and finally clarified some problems in the mechanism.
In reference section you should adopt and correct references according to the rules of the journal Animals, like this: “Mota, P.C.; Ehmcke, J.; Westernströer, B.; Gassei, K.; Ramalho-Santos, J.; Schlatt, S. Effects of different storage protocols on cat testis tissue potential for xenografting and recovery of spermatogenesis. Theriogenology 2012, 77, 299–310.”
Correction: We are very grateful to the reviewers for their suggestions on the reference format of this article, so that our article meets the publication requirements. We have made changes in the article.
Conclusion need major revision with referring to specific genes with the most differences found in experiments.
Correction: We thank the reviewer for useful suggestions for the refinement of our results, and we have made changes to the conclusion in the article. The changes are as follows.
Among them, genes such as ALPL, FABP3, VCAN and SLC11A1 have been highly ex-pressed at different times and ALOX15, SNCG, EPDR1, CNN1 were all down-regulated compared with normal dairy goats. At the same time, this study also found some interesting problems. When we analyzed the data, we found that the mRNA level and protein expression level of some data were opposite. In the 72h infection group, mRNA expression was up-regulated and protein expression was down-regulated in 7 groups, TNXB, NFIA, PROC, SPIN1, NR2C2 nuclear receptor sub-family 2 group C member 2, Carboxypeptidase D, ARHGAP11B were included. This phenomenon may be related to the epigenetic modification of mRNA and protein.
Specific comments
You could name three testing group with symbols, for example: group A (72 hours), group B (144 hours) and group C injected with normal saline (control).
Correction: Thank you very much for the reviewer's suggestion, this does make our group clear in the article, but we think it is easy to be confused with AB in the picture, so we decided to take 72h group, 144h group and NC group to represent our groups.
Corynebacterium pseudotuberculosis or C. pseudotuberculosis should be in italic as Latin species name in every part of article.
Correction: We thank the reviewer for their reasonable comments on the detailed part of the article, and we have revised all required italics in the article.
“And most of the sheep raising areas in our country have the disease”, please cite the name of your country.
Correction: We thank the reviewer for useful suggestions. “and most of the sheep raising areas in our country have the disease” should be changed to “and most of the sheep raising areas in Chia have the disease”.
In Materials and Methods section, In Figure 1. You wrote C. Pseudotuberculosis with uppercase instead of C. pseudotuberculosis.
Correction: We thank the reviewer for their reasonable comments on the detailed part of the article, and we have corrected the error in the picture.
Reviewer 4 Report
This is an article that has scientific merit, but I believe that it needs a contribution from a statistician because the sperman correlation method would not be the most appropriate method to indicate an association between the transcriptome and the proteome, that is, there is a cause and effect relationship?
Author Response
Reviewer 4:
This is an article that has scientific merit, but I believe that it needs a contribution from a statistician because the spearman correlation method would not be the most appropriate method to indicate an association between the transcriptome and the proteome, that is, there is cause and effect relationship?
Explanation: First of all, I would like to thank the reviewer for affirmations on most of the content of this article. Then we attached great importance to the problem of reviewers, and have consulted a large number of relevant literatures to understand whether spearman analysis can be applied to transcriptome and proteome analysis. Here are some of the content we found and hoped to share with the reviewer. Wikipedia Definition: In statistics, Spearman’s rank correlation coefficient or Spearman’s ρ, named after Charles Spearman is a nonparametric measure of rank correlation (statistical dependence between the rankings of two variables). It assesses how well the relationship between two variables can be described using a monotonic function. Correlation is the degree to which two variables are linearly related. Broadly speaking, a correlation is actually any statistical relationship between two random variables in bivariate data, whether causal or not.
Meanwhile, we found that Spearman analysis was used in several articles for transcriptome and proteome analysis. For example, Integrated proteotranscriptomics of breast cancer reveals globally increased protein-RNA concordance associated with subtypes and survival, Proteomic and transcriptomic profiling reveal different aspects of aging in the kidney and Proteome and Transcriptome Analysis of Gonads Reveals Intersex in Gigantidas haimaensis etc.
Finally, we would like to thank the reviewer again for their valuable comments, which allowed us to have a deeper understanding of statistical knowledge and made the article more complete.
Round 2
Reviewer 4 Report
The Spearman rank order correlation is the nonparametric version of the Pearson product moment correlation. The Spearman correlation coefficient, (ρ, also represented by rs) measures the strength and direction of the association between two classified variables.
You need two variables that are ordinal, interval, or ratio. Although you would normally expect to use a Pearson product-moment correlation on interval or ratio data, Spearman correlation can be used when Pearson correlation assumptions are markedly violated. However, Spearman's correlation determines the strength and direction of the monotonic relationship between your two variables (A monotonic relationship is a relationship that does one of the following: (1) as the value of one variable increases the value of the other variable also increases; or (2) as the value of one variable increases, the value of the other variable decreases), rather than the strength and direction of the linear relationship between its two variables, which is what Pearson's correlation determines.
The question I ask the authors is: have the assumptions of Pearson's correlation been violated? Wouldn't it be interesting to explore this linear relationship with this type of correlation data between protein and mRNA abundance? And perhaps better to explore this linear relationship through a linear regression in quantifying this relationship?
Author Response
Thank you very much for the comments made by the reviewer. First of all, we refer to a large amount of data and make the following explanations for the three questions of the reviewer. Firstly, the assumption of Pearson correlation mentioned that the variables must be continuous, there must be a linear relationship between the variables, and the two variables must also conform to a bivariate normal distribution. In this study, transcriptome and proteome data are not necessarily continuous variables. Due to the presence of epigenetic modifications in mRNA and protein, there is not necessarily a linear relationship between them, and transcriptome and proteome data do not necessarily conform to the normal distribution, so Pearson's correlation is not suitable for this study. For the second and third questions, our interpretation is that the relationship between mRNA and protein is mostly linear, but due to the presence of epigenetic modifications, such as methylation, phosphorylation and acetylation, a small proportion of mRNAs and proteins do not necessarily have a linear relationship.
Then we consulted the data and used Spearman's correlation analysis to meet two conditions: variables that were not normally distributed (or had outliers that could not be removed). There is a monotonic relationship between the variables. This is consistent with the relationship between changes in mRNAs and proteins abundance. This study also only explored the strength and direction of the association between mRNAs and proteins abundance variables.